# Methyl Salicylate Increases Attraction and Function of Beneficial Arthropods in Cranberries

**DOI:** 10.3390/insects10120423

**Published:** 2019-11-25

**Authors:** Jordano Salamanca, Brígida Souza, Vera Kyryczenko-Roth, Cesar Rodriguez-Saona

**Affiliations:** 1Escuela de Ciencias Agrícolas, Pecuarias y del Medio Ambiente (ECAPMA), Universidad Nacional Abierta y a Distancia (UNAD), Bogotá 110111, Colombia; 2Departamento de Entomologia, Universidade Federal de Lavras, Lavras 37200-000, Minas Gerais, Brasil; brgsouza@den.ufla.br; 3P.E. Marucci Center for Blueberry & Cranberry Research, Rutgers University, Chatsworth, NJ 08019, USA; vera.kyryczenko@rutgers.edu (V.K.-R.); crodriguez@njaes.rutgers.edu (C.R.-S.)

**Keywords:** *Vaccinium macrocarpon*, cranberries, HIPVs, natural enemies, predation, biological control

## Abstract

Methyl salicylate (MeSA) is an herbivore-induced plant volatile (HIPV) known to attract the natural enemies of herbivores in agro-ecosystems; however, whether this attraction leads to an increase in natural enemy functioning, i.e., predation, remains largely unknown. Here, we monitored for 2 years (2011–2012) the response of herbivores and natural enemies to MeSA lures (PredaLure) by using sticky and pitfall traps in cranberry bogs. In addition, European corn borer, *Ostrinia nubilalis*, egg masses were used to determine whether natural enemy attraction to MeSA leads to higher predation. In both years, MeSA increased adult hoverfly captures on sticky traps and augmented predation of *O. nubilalis* eggs. However, MeSA also attracted more phytophagous thrips and, in 2012, more plant bugs (Miridae) to sticky traps. Furthermore, we used surveillance cameras to record the identity of natural enemies attracted to MeSA and measure their predation rate. Video recordings showed that MeSA lures increase visitation by adult lady beetles, adult hoverflies, and predatory mites to sentinel eggs, and predation of these eggs doubled compared to no-lure controls. Our data indicate that MeSA lures increase predator attraction, resulting in increased predation; thus, we provide evidence that attraction to HIPVs can increase natural enemy functioning in an agro-ecosystem.

## 1. Introduction

Biological control services provided by natural enemies of herbivores constitute an integral part of integrated pest management (IPM) programs in agricultural crops [1,2,3,4]; however, these services can be negatively impacted by certain anthropogenic activities [5]. For instance, humans have often domesticated crops for traits associated with high productivity as opposed to defenses against herbivory. As a result, plant defenses are often lower in domesticated plants than their wild ancestors [6,7,8]. For example, crop domestication can inadvertently reduce the emissions of herbivore-induced plant volatiles (HIPVs), as has been observed in cranberry, *Vaccinium macrocarpon* Ait. (Ericaceae) [9], as well as in other crops (e.g., [10,11,12]; but see [13]).

Elevated HIPV production and emission are often triggered by herbivore feeding and oviposition [14,15]. These HIPVs are considered a form of indirect defense in plants because they serve as important foraging cues for natural enemies of herbivores—such as insect predators—during prey location [16,17,18,19,20,21] and could, thus, enhance natural enemy ecosystem function (i.e., predation). For example, Kessler and Baldwin [22] demonstrated that HIPVs emitted from wild tobacco plants—*Nicotiana attenuata* Torr. ex S. Watson—increase egg predation rates by a generalist predator in nature. In addition to being more detectable, HIPVs represent a more reliable cue than volatiles from undamaged plants for foraging natural enemies because they are usually associated with the presence of prey [17,23]. However, whether biological control services can be restored in agro-ecosystems through the manipulation of HIPVs to increase the attraction of beneficial arthropods remains an open question [24,25,26,27].

In recent decades, several studies have tested synthetic versions of HIPVs to attract natural enemies of herbivorous pests in agricultural systems (e.g., [28,29,30]). Methyl salicylate (MeSA) is an ubiquitous HIPV [31] that is often induced by feeding by herbivores from different feeding guilds—including mites [32], aphids [33,34], and beetles [35]—and has been proven to be attractive to various beneficial arthropods, including insect predators, in agro-ecosystems [9]. For example, James and Price [36] reported the attraction of several insect predators—including *Chrysopa nigricornis* Burmeister, *Hemerobius* sp., *Deraeocoris brevis* (Uhler), *Stethorus punctum picipes* (Casey), and *Orius tristicolor* (White)—to MeSA in grapes and hops. Similarly, the predatory seven-spotted lady beetle *Coccinella septempunctata* L. was attracted to MeSA in soybean fields [33]. Currently, a MeSA-containing lure named PredaLure (AgBio Inc., Westminster, CO, USA) is commercially available for the attraction of natural enemies. Previously, Lee [37] reported the attraction of chrysopids and *O. tristicolor* to PredaLure in strawberries. In a previous study to investigate the response of predatory insects to MeSA in cranberries, we found greater numbers of adult hoverflies, lady beetles, and green lacewings in PredaLure-baited sticky traps than in unbaited traps [9].

Thus far, most studies using synthetic HIPVs have focused on the attraction of natural enemies to baited lures (e.g., [9,33,36,37]); however, only a few studies have investigated whether this attraction leads to greater natural enemy function, i.e., increased predation [38,39]. Moreover, these previous studies are often conducted in a single year or for a short period during the growing season (e.g., [9,30,33,36]). Therefore, in this study, we tested the hypotheses that HIPVs provide multi-year, season-long attraction of natural enemies of herbivorous pests in an agricultural crop—cranberries—and whether the attraction of predators to HIPVs leads to increased predation. To this end, we conducted a series of field studies in 2011, 2012, and 2014 with MeSA (PredaLure) as our predator-attractive HIPV to (1) determine the multi-year (2011–2012), season-long (i.e., from bud break/leaf elongation to fruit maturation) response of air-borne and soil-inhabiting beneficial (i.e., predator and parasitoids) and antagonistic (i.e., herbivores) arthropod communities to MeSA in cranberries; (2) measure the effects of MeSA on predation of sentinel eggs to assess the ecosystem function provided by predators; and (3) record the identity of predators attracted to MeSA through the use of surveillance cameras and measure their ecosystem function.

## 2. Materials and Methods

### 2.1. Arthropod Community Response to MeSA

This 2-year (2011–2012) season-long (May through August) experiment tested the response of natural enemies and phytophagous arthropods to the MeSA-baited lure PredaLure in cranberry—*V. macrocarpon*—fields (commonly referred to as bogs or marshes because of the use of water for harvesting and overwinter protection).

The experiment was conducted in eight cranberry bogs (1 to 3 ha each) with cvs. ‘Stevens’ or ‘Early Black’ located in four commercial farms (two bogs per farm) in Chatsworth, New Jersey (USA). Stevens and Early Black are the two most popular cultivars in New Jersey, planted in >70% of the cranberry acreage. To monitor flying predators, parasitoids, and herbivores, we followed similar methods as described in Rodriguez-Saona et al. [9]. A pair of yellow sticky traps (23 × 28 cm, no-bait Pherocon AM; Trécé Inc., Adair, OK, USA) was placed in each bog. One of the traps was baited with MeSA (PredaLure; 5 g load/lure; 90-day lure; average release rate ~35 mg/day over a 4-week period at 30 °C constant in the lab; AgBio Inc.) while the other trap had no lure (unbaited control). Traps were placed at least 100 m apart within a bog to minimize treatment overlap [9,37] and were attached vertically to 40-cm-tall metal poles such that the trap bottoms were ~10 cm aboveground, i.e., just above canopy level. MeSA lures were attached to the poles so that they were adjacent, but not touching, the traps and were ~20 to 30 cm aboveground. In addition, we monitored ground-dwelling predators and herbivores by using pitfall traps placed next to each metal pole (1 trap per pole). Pitfall traps consisted of plastic cups (10.5 × 8 cm) buried in the soil such that the cup rim was level with the soil surface; traps were filled halfway with a solution of propylene glycol and water (70:30) and covered with a small wooden board (to protect from rain and falling leaves) raised 2 to 3 cm above the trap with nails in each corner.

The study ran for a total of ~4 months from 20 May (bud break and leaf elongation) through to 26 August 2011 (fruit maturation) and from 18 May through to 13 August 2012. The position of each treatment within a bog was assigned randomly. Every 2 weeks, sticky and pitfall traps were collected, brought to the laboratory, and replaced. MeSA lures (PredaLures) were replaced every 4 weeks, as recommended by the manufacturer. In the laboratory, numbers of adult insect predators on traps were counted under a stereomicroscope (Nikon SMZ-U, Tokyo, Japan). Common arthropods (predators, parasitoids, and herbivores) collected from traps were identified to family and, when possible, to species or genus. All bogs were managed under standard pest management guidelines [40] and received minimal insecticide input (i.e., ≤3 applications throughout the season, either before and/or soon after bloom) (Appendix A).

### 2.2. Field Egg Predation

Concurrently with our trapping experiment (above), we conducted a field experiment in 2011 and 2012 to study whether the attraction of beneficial arthropods to MeSA increases their ecosystem function (i.e., sentinel egg predation) in cranberries. For this, eight cranberry bogs (1 to 3 ha each; cvs. Stevens or Early Black) were selected from the same four commercial farms mentioned above (two bogs per farm) but were different from those used in the predator attraction experiment. In each bog, we placed two metal poles at least 100 m apart, as described above. A MeSA lure (PredaLure) was attached to one of the metal poles while the other pole contained white cardboard of a similar size as the PredaLure (no-lure control).

To measure egg predation, we used sentinel egg masses of the European corn borer *Ostrinia nubilalis* (Hübner) (Lepidoptera: Crambidae) obtained from Rincon-Vitova Insectaries (Ventura, CA, USA). We used *O. nubilalis* eggs because (1) they are commercially available and, hence, large quantities could be purchased as needed; (2) these eggs are readily utilized as food by several arthropod predators (e.g., [41,42,43]), including those found in cranberry bogs in the USA; and (3) *O. nubilalis* larvae do not feed on cranberries, thus preventing crop injury in case some eggs accidently hatched in the field. A piece of wax paper containing an egg mass was stapled to 15-cm-tall wooden stakes (plant labels; Gempler’s, Belleville, WI, USA), allowing for each egg mass to be ~5 to 10 cm above the cranberry canopy. Ten stakes with egg masses were placed within 20 cm from each metal pole (N = 80 egg masses per treatment per deployment date). Each egg mass contained 25.5 ± 11 eggs.

Sentinel egg masses were deployed in cranberry bogs eight times in both years (on 19 May, 2 June, 15 June, 30 June, 14 July, 28 July, 11 August, and 25 August 2011; and on 17 May, 1 June, 14 June, 28 June, 12 July, 26 July, 9 August, and 16 August 2012). Eggs were deployed within 24 h of arrival, left in the bogs for 24 to 48 h, and then returned to the laboratory. The number of eggs per mass was recorded before and after each deployment; eggs were examined under the microscope for signs of predation after being retrieved from the field.

### 2.3. Video Recordings

A field experiment was conducted to determine the identity of arthropod predators attracted to MeSA and their effect on egg predation in cranberries. In a cranberry bog at the Rutgers P.E. Marucci Center (Chatsworth, NJ), we placed eight video cameras (Night Owl, http://nightowlsp.com/) that were connected to an 8-channel DVR Surveillance System (Q-See, Anaheim, CA, USA). The cameras were arranged around a circle with an approximately 8-m radius (Figure 1). Four video cameras were placed close to a station containing a 15-cm-tall wooden stake (Gempler’s) that had a green cardboard piece (2 × 2 m) with five *O. nubilalis* sentinel egg masses glued onto it next (~10 cm) to a similar stake with a MeSA lure (PredaLure), while the other four cameras were placed close to a station containing a wooden stake with the sentinel eggs and a stake without a MeSA lure, which was replaced by a white cardboard piece of a similar size as the PredaLure (Figure 1). MeSA-baited and unbaited stations were alternated with each other, and stations were separated by ~6.5 m. Video cameras were positioned so that insects visiting the cardboards could be recorded and were powered by two 12-V lead-acid E-bike batteries with a capacity of 24 h each, which allowed at least 48 h of continuous recording.

Video recordings were conducted on 2 consecutive days, twice a week, from May through to August of 2014, for a total of 14 different recording periods and a total of 576 h recorded (~41 h/recording). After collection, the video footage was transferred to a portable hard-drive and stored until viewed on a computer. To measure predator visitation to each cardboard, we played back each recoding (pixel resolution of 720 × 576) and counted the number of predator visitations. All predators were identified to the lowest taxonomic level possible given the resolution of the video. In addition, to determine egg predation, we counted the number of eggs per mass before and after each deployment and examined eggs under the microscope for signs of predation after being retrieved from the field.

### 2.4. Data Analyses

All analyses were conducted in R 3.3.1 [44]. Prior to analyses, all data were first checked for normality by using the Shapiro–Wilk test [45] and for homoscedasticity by using Levene’s test (‘car’ package in R). First, we analyzed the response of arthropods to MeSA at the community level. For this, arthropods captured on yellow sticky traps and pitfall traps were classified based on feeding groups into predators, parasitoids, and herbivores and were analyzed separately. We then used multivariate analyses (two-way multivariate analysis of variance (MANOVA)) to test the effects of ‘Treatment’ (−MeSA or +MeSA), ‘Date’, and their interaction on the abundance of each group, i.e., predators, parasitoids, and herbivores. Additionally, we used principal component analysis (PCA) to visualize differences between treatments for arthropod communities significantly affected by MeSA based on MANOVA. PCA score and loading plots were drawn in R using the ‘ggplot2′ package [46].

If MANOVAs were significant for treatment effects, we used univariate statistics (two-way analysis of variance (ANOVA)) to determine which taxon within each of the communities was significantly affected by MeSA. The model included treatment (−MeSA or +MeSA), date, and their interaction as independent variables. Mean differences were compared using Tukey honestly significant difference (HSD) test (*p* ≤ 0.05) after a significant *F* test (agricolae package in R). Similarly, the effects of treatment (−MeSA or +MeSA), date, and their interaction on percent sentinel egg predation were analyzed by two-way ANOVA, followed by Tukey HSD test (*p* ≤ 0.05). Finally, we tested the effect of treatment (presence or absence of MeSA) on the percent number of visits by natural enemies from the camera recordings and on the percent egg predation using chi-square (*χ*^2^) tests.

If needed, data were transformed prior to MANOVA and ANOVA analyses by using ln(*x* + 0.5) to meet assumptions of normality. Percentages of the number of visits and egg predation were arcsine square-root transformed. Untransformed data are presented in figures.

## 3. Results

### 3.1. Arthropod Community Response to MeSA

In 2011 and 2012, treatment (MeSA) and date, but not their interaction, had a significant effect on the predatory arthropod community captured on sticky traps (Table 1). In 2012, treatment (MeSA) and date also had a significant effect on the herbivorous arthropod community captured on sticky traps. MeSA had no effects on the parasitoid community captured on sticky traps in 2011 and 2012 or on the herbivore community captured on sticky traps in 2011 (Table 1). Additionally, MeSA had no effect on the predatory arthropod community captured in pitfall traps in both years (Table 1) or on adult scarab (Scarabaeidae) and soldier (Cantharidae) beetles captured in pitfall traps in 2011 and 2012, respectively (*p* > 0.05 for both).

The PCA for 2011 showed a distinct composition of the natural enemy communities according to the MeSA treatment, with the first two components explaining ~73% of the variance (Figure 2A). The first PC explained 47% of the variation and clearly separated the MeSA and control treatments, while the second PC explained 26% of the variation. In 2012, the first two PCs explained ~75% of the variation of natural enemy communities between treatments (Figure 2B). The first PC explained 47% of the variation while the second PC explained 27% of the variation and separated the MeSA and control treatments. In both years, sticky traps baited with MeSA captured greater numbers of adult hoverflies (Syrphidae) than unbaited traps (Table 2; Figure 3A,C). On average, MeSA-baited traps captured approximately three times more hoverflies than unbaited traps (Figure 3B,D). The most common hoverfly on traps was *Toxomerus marginatus* (Say).

In 2012, the first two PCs explained ~72% of the variation of herbivore communities between treatments (Figure 2C). The first PC explained 46% of the variation while the second PC explained 26% of the variation and separated the MeSA and control treatments. Plant bugs (Miridae) were approximately five times more abundant on sticky traps baited with MeSA than unbaited traps (Figure 3C,E). In addition, the number of phytophagous thrips on MeSA-baited sticky traps was 22% and 33% higher than on unbaited traps in 2011 and 2012, respectively (2011: MeSA-baited trap (mean ± SE) = 2409 ± 189, unbaited trap = 1966 ± 172, *F* = 4.15, df = 1, 103, *p* = 0.04; 2012: MeSA-baited trap = 702.25 ± 68.98, unbaited trap = 528.98 ± 52.19, *F* = 3.21, df = 1, 94, *p* = 0.07).

### 3.2. Field Egg Predation

In 2011, percent predation of *O. nubilalis* eggs was 3% higher near the MeSA lures (PredaLures) than without the lures (significant treatment effect; *F* = 4.81, df = 1, 1123, *p* = 0.02) (Figure 4A,B). There was also an effect of date (*F* = 17.77, df = 7, 1123, *p* < 0.001) but not a treatment × date interaction (*F* = 0.69, df = 7, 1123, *p* = 0.67).

In 2012, percent predation of *O. nubilalis* eggs was 6% higher near the MeSA lures (PredaLures) than without the lures (significant treatment effect; *F* = 14.12, df = 1, 1180, *p* < 0.001) (Figure 4C,D). There was also an effect of date (*F* = 14.91, df = 7, 1180, *p* < 0.001) but not a treatment × date interaction (*F* = 1.23, df = 7, 1180, *p* = 0.28).

### 3.3. Video Recordings

In total, we observed 94 visits of natural enemies to the cardboard pieces containing sentinel eggs. Out of these visits, 40 (42.6%) were by adult lady beetles (Coccinellidae), 20 (21.3%) by adult hoverflies (Syrphidae), 9 (9.6%) by spiders (Araneae), 9 (9.6%) by predatory mites (Acari: Phytoseiidae), and 16 (17%) by other arthropods. There were 10%, 15%, and 8% more lady beetles (χ^2^ = 3.71, df = 1, *p* = 0.05), hoverflies (χ^2^ = 9, df = 1, *p* < 0.01), and predatory mites (χ^2^ = 7.75, df = 1, *p* < 0.01) that visited the MeSA-baited stations than the unbaited stations (Figure 5A,B,D, respectively). In contrast, spiders visited the unbaited stations more often than the MeSA-baited stations (χ^2^ = 15.27, df = 1, *p* < 0.001; Figure 5C).

Percent predation of *O. nubilalis* eggs was approximately two times higher in MeSA-baited stations than in unbaited stations (MeSA-baited station (mean ± SE) = 23.37% ± 2.74%, unbaited station = 11.92% ± 2.05%; χ^2^ = 3.71, df = 1, *p* = 0.05).

## 4. Discussion

From the field trapping and video recording experiments, we demonstrated that MeSA attracts the natural enemies of herbivores—including adult hoverflies, adult lady beetles, and predatory mites—in cranberries and that this attraction led to higher predation of sentinel eggs. However, MeSA also attracted insect herbivores—such as thrips and plant bugs (Miridae)—and repelled spiders, suggesting potential ecological costs of HIPVs.

Across 2 years and in both our field trapping and video recording experiments, MeSA baits consistently attracted adult hoverflies in cranberry bogs. This agrees with our previous studies [9,47]. For instance, in a short-term study (1 year, 1.5 months), we showed high attraction of syrphid flies—*T. marginatus*—to PredaLure-baited sticky traps [9]. Similarly, De Lange et al. [47] compared the attraction of various HIPVs alone and in combination to natural enemies and found that MeSA, but not (*Z*)-3-hexenyl acetate, linalool, or β-caryophyllene, attracted syrphids in cranberries. The present study complements these previous findings by showing a multi-year, season-long attraction of syrphids in cranberries. Although much is still widely unknown, syrphids can potentially play two important ecosystem services in cranberries—the immatures can serve as biological control agents of insect pests like cranberry tipworm, *Dasineura oxycoccana* (Johnson), larvae [48], while the adults can be pollinators [49]. Previous studies have also reported the attraction of syrphids to MeSA in other crops like hops [28] and soybean [38], but Yu et al. [29] found no response of the syrphid fly *Epistrophe balteata* (De Geer) to MeSA in cotton, indicating a variation in response depending on the crop and syrphid species.

In addition to syrphids, video recordings showed the attraction of adult lady beetles and predatory mites to MeSA in cranberries. From our previous studies, the response of adult lady beetles to MeSA tends to be inconsistent. While Rodriguez-Saona et al. [9] showed attraction of lady beetles to MeSA, De Lange et al. [47] showed no response. In this study and in the study by De Lange et al. [47], field experiments were done in commercial farms under standard pest management—which may reduce lady beetle density—whereas our video recordings here and the study by Rodriguez-Saona et al. [9] were done in bogs that had not received insecticide applications for a few or several years prior to the experiment, which allowed for high lady beetle densities. Thus, it is likely that the effects of MeSA on lady beetles is density dependent. Also, spatial and temporal variation in the lady beetle species composition could affect their response to MeSA. Attraction of predatory mites to MeSA has been widely reported under laboratory conditions [50,51]. Predatory mites were attracted to MeSA in video recordings but not in field trapping experiments—possibly, colored sticky and pitfall traps are not appropriate devices for capturing these mites. The use of video cameras can thus unravel information missed by other surveillance methods. Similar to Lee [37], we found no response of ground-dwelling predators collected in pitfall traps, such as spiders (Araneae) and carabid and staphylinid beetles, to MeSA.

Attraction of predators to HIPVs may result in increased ecosystem function. In fact, in both field experiments (trapping and video recordings), we observed higher predation of sentinel eggs in locations adjacent to MeSA lures (PredaLure) than in no-lure control sites. To date, most field studies on the effects of HIPVs on natural enemies have documented attraction only; in contrast, there are only a few examples on whether this attraction leads to an increase in predation or parasitism (i.e., natural enemy function). For example, in cotton fields, parasitism of *Lygus lineolaris* (Palisot de Beauvois) eggs by the mymarid *Anaphes iole* Girault was higher when the eggs were associated with (*Z*)-3-hexenyl acetate or α-farnesene than with no-lure controls [52]. MeSA lures also reduced the abundance of soybean aphids in organic soybean plots [38], whereas lures baited with isolates of ocimene and farnesene increased parasitism of the asparagus miner *Ophiomyia simplex* (Loew) by pteromalid wasps [53]. Yet, this is the first study to document that natural enemy attraction to MeSA may lead to greater predation in cranberries. Although adult hoverflies were consistently attracted to MeSA, it is unlikely that they contributed to the observed increase in egg predation because they are pollen and nectar feeders [54]. In fact, in both years of our study, their peak activity coincided with bloom (see also [9]), whereas their predaceous larvae were never seen near the sentinel eggs. In contrast, at least from the video recording experiment, higher egg predation could be attributed to greater adult lady beetle and predatory mite visitation to MeSA-baited stations. Nevertheless, the identity of predators responsible for preying on eggs in commercial cranberry bogs remains largely unknown. Our study used eggs of a non-cranberry pest (i.e., the European corn borer, *O. nubilalis*) to prevent infestation of a pest in commercial bogs. Further studies are needed to determine if natural enemy attraction to MeSA can reduce populations of a cranberry pest and whether this results in lower crop damage and an increase in yield. Moreover, the attraction of multiple predators can lead to intraguild predation (e.g., [55]), which could dampen biological control [56]. Our data, however, do not show interference among predators, as this attraction resulted in greater egg predation.

A drawback of using HIPVs in agro-ecosystems is the potential attraction of the pests themselves [57]. Although MeSA attracted thrips and plant bugs in cranberries, these insects are not considered of economic importance. Direct feeding by thrips is not known to cause any significant crop injury, and they have yet to be conclusively implicated in disease transmission, such as tobacco streak virus, in cranberries [58]. On the other hand, a plant bug, the mirid *Plagiognathus repetitus* (Knight), has been reported to reduce cranberry yields [59]. Mirids are omnivorous piercing-sucking insects that can also prey on eggs [55] and could thus be partially responsible for preying on the sentinel eggs in our study. Another ecological cost that HIPVs could incur is by repelling certain natural enemies. Although the repellency of natural enemies by HIPVs has rarely been reported (but see [60]), in the present study, we found some evidence that spiders could be negatively affected by MeSA. De Lange et al. [47] also reported lower captures of parasitic megaspilid wasps in MeSA-baited traps than in unbaited traps in cranberries. Whether or not these unintended, non-target consequences of HIPVs on herbivores and natural enemies interfere with biological control, resulting in yield losses, needs investigation.

Besides their use to attract natural enemies to crops, which may come with some risks [57], HIPVs can potentially serve as a pest management decision tool for farmers for conservation biological control, as has been proposed by Jones et al. [61]. In fact, once the functionality of keystone natural enemies has been established and as long as their populations are large enough that their removal does not disrupt biological control, HIPVs, such as MeSA, could be used as a monitoring tool to assess population fluctuations of these natural enemies within and among growing seasons in cropping systems (e.g., [62,63]). For instance, yellow sticky traps baited with MeSA can be used to monitor seasonal fluctuations of syrphid populations in cranberries. This information can then be used to conserve this and other biological control agents, for example, by limiting insecticide use or using reduced-risk/low-toxicity insecticides during their peak seasonal activity and, therefore, mitigating the non-target, negative impacts of insecticide applications on biological control services in agro-ecosystems [64].

## 5. Conclusions

This study provides evidence that the attraction of natural enemies to an HIPV, such as MeSA, can enhance their function by increasing predation in an agro-ecosystem. However, it also raises possible concerns of using HIPVs, such as the attraction of herbivores or repellency of potential bio-control agents. Moreover, our study highlights the importance of using multiple surveillance methods (i.e., yellow sticky and pitfall traps and video cameras), as they can provide a more comprehensive view of the community-level effects of HIPVs. Lastly, the present study was designed as a proof of concept to answer whether the attraction of natural enemies to HIPVs could lead to increased predation under field conditions. Additional studies are needed to investigate what would happen to predators and their services if prey items are absent or not available at close proximity to MeSA lures.

## Figures and Tables

**Figure 1 insects-10-00423-f001:**
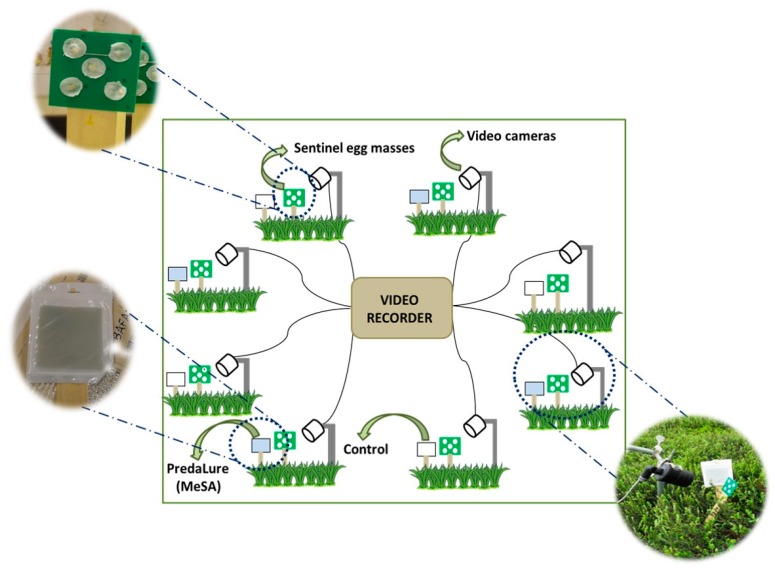
Video recording set-up. Eight video cameras were connected to an eight-channel DVR Surveillance System (“Video Recorder”) and arranged around a (8-m radius) circle. Baited stations consisted of a wooden stake with five *Ostrinia nubilalis* egg masses on a cardboard next to a wooden stake containing a methyl salicylate (MeSA) lure (PredaLure); unbaited stations were similar to the baited stations but without the MeSA lure (control). Four video cameras faced a baited station and four cameras faced an unbaited station; stations were separated by ~6.5 m.

**Figure 2 insects-10-00423-f002:**
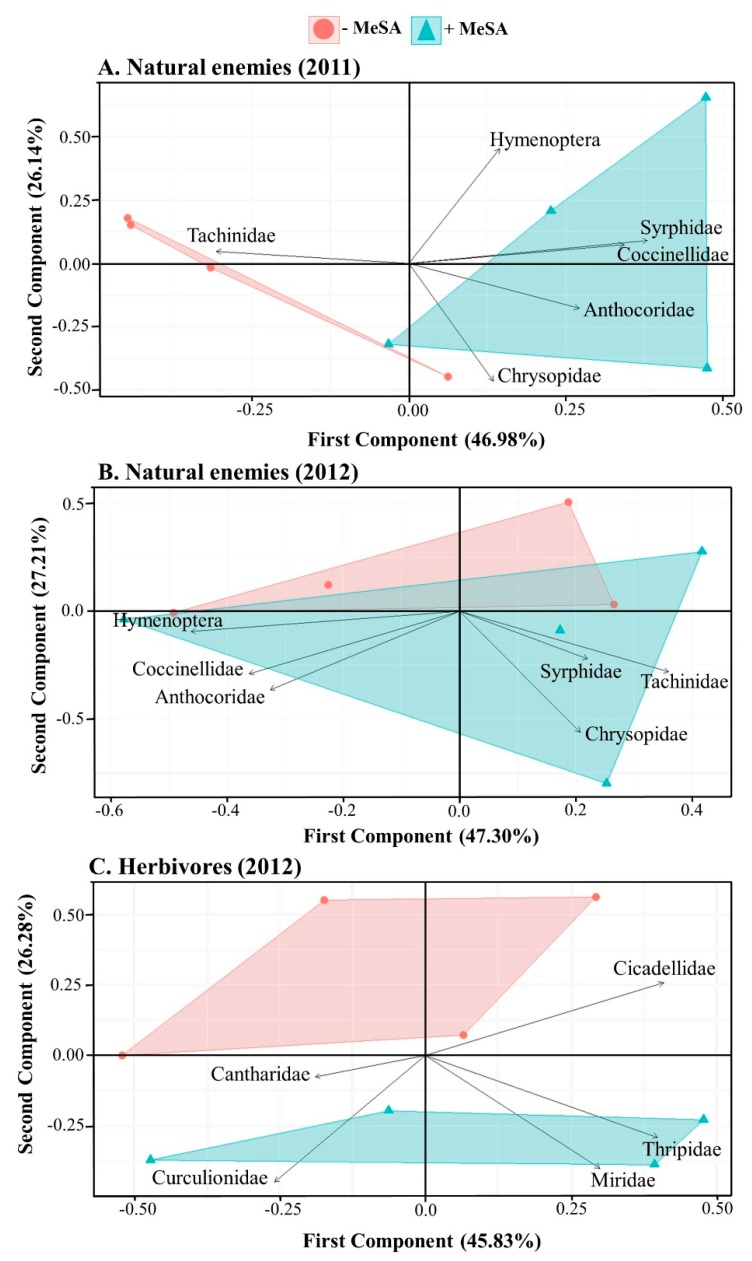
Principal component analysis (PCA) score and loading plots (first and second PCs) on the effects of control (circles) and methyl salicylate (MeSA; PredaLure) (triangles) on natural enemies in 2011 (**A**) and 2012 (**B**) and on herbivores (**C**) in 2012.

**Figure 3 insects-10-00423-f003:**
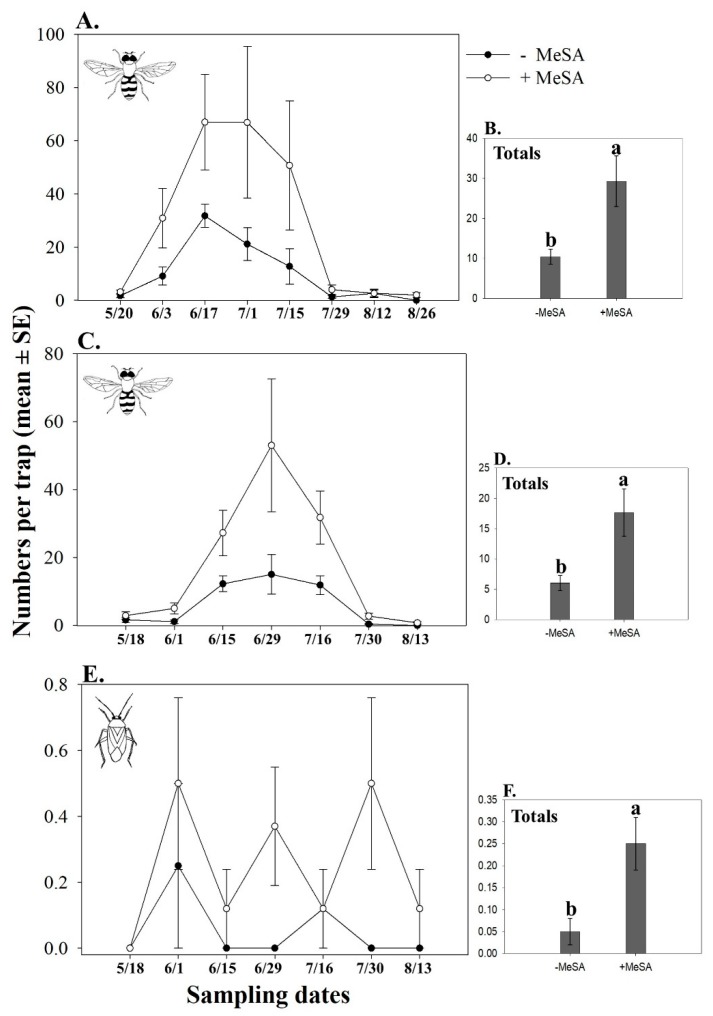
Seasonal mean (± SE) (**A**,**C**) and total (±SE) numbers (**B**,**D**) of adult hoverflies, mainly *Toxomerus marginatus* (Syrphidae), captured on sticky traps in 2011 (**A**,**B**) and 2012 (**C**,**D**). Seasonal mean (±SE) (**E**) and total (±SE) numbers (**F**) of plants bugs (Miridae) captured on sticky traps in 2012 in cranberry bogs. Sticky traps were either unbaited (control) or baited with methyl salicylate (MeSA; PredaLure). Sampling dates = month/day. Different letters indicate significant differences among treatments (*p* ≤ 0.05).

**Figure 4 insects-10-00423-f004:**
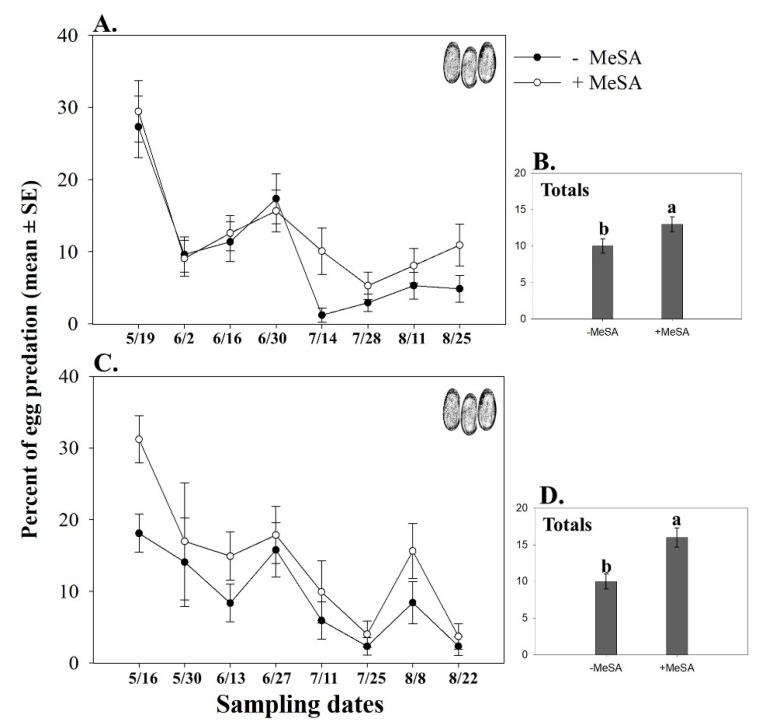
Seasonal mean (±SE) (**A**,**C**) and total (±SE) percent predation (**B**,**D**) of *Ostrinia nubilalis* eggs in 2011 (**A**,**B**) and 2012 (**C**,**D**) in cranberry bogs. Eggs were either near a pole with a lure baited with methyl salicylate (MeSA; PredaLure) or an empty pole. Sampling dates = month/day. Different letters indicate significant differences among treatments (*p* ≤ 0.05).

**Figure 5 insects-10-00423-f005:**
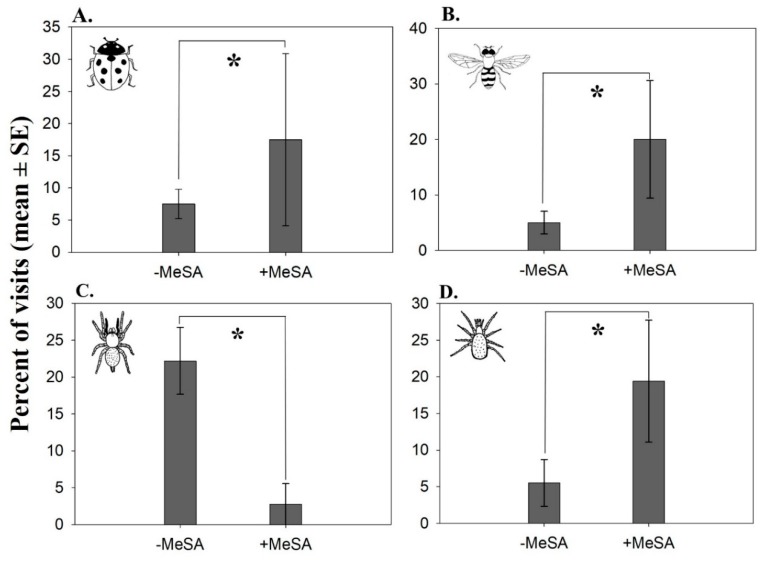
Mean (± SE) percent of visits by adult lady beetles (Coccinellidae) (**A**), adult hoverflies—mainly *Toxomerus marginatus*—(Syrphidae) (**B**), spiders (Araneae) (**C**), and predatory mites (Acari: Phytoseiidae) (**D**) to methyl salicylate (MeSA; PredaLure)-baited and unbaited stations. Each station consisted of a wooden stake with a green cardboard piece (2 × 2 m) containing five *O. nubilalis* sentinel egg masses (see Figure 1). Video recordings were done in 2014 in a cranberry bog. An asterisk (*) indicates significant differences among treatments (*p* ≤ 0.05).

**Table 1 insects-10-00423-t001:** Results of MANOVA for the effects of methyl salicylate (MeSA) (‘Treatment’), sampling date (‘Date’), ‘Treatment × Date’ interaction, and block on the attraction of arthropod predators, parasitoids, and herbivores to yellow sticky traps and of predators to pitfall traps in 2011 and 2012 in commercial cranberry bogs.

Trap Type	Years	Guilds	Variables															
			Treatment				Date				Treatment × Date				Block			
			Wilk’s λ	*F*	*df* ^a^	*P* ^b^	Wilk’s λ	*F*	*df* ^a^	*P* ^b^	Wilk’s λ	*F*	*df* ^a^	*P* ^b^	Wilk’s λ	*F*	*df* ^a^	*P* ^b^
Sticky	2011	Predators ^c^	0.81	5.54	1, 104	**<0.001**	0.18	7.8	7, 104	**<0.001**	0.85	0.56	7, 104	0.96	0.98	0.5	1, 104	0.73
		Parasitoids ^d^	0.95	2.64	1, 104	0.07	0.5	5.93	7, 104	**<0.001**	0.86	1.14	7, 104	0.32	0.97	1.39	1, 104	0.25
		Herbivores ^e^	0.94	1.25	1, 103	0.28	0.2	5.56	7, 103	**<0.001**	0.69	1.06	7, 103	0.37	0.93	1.46	1, 103	0.2
	2012	Predators ^c^	0.71	9.07	1, 94	**<0.001**	0.16	0.97	6, 94	**<0.001**	0.83	0.7	6, 94	0.84	0.9	2.46	1, 94	**0.05**
		Parasitoids ^d^	0.99	0.27	1, 94	0.76	0.44	7.79	6, 94	**<0.001**	0.9	0.8	6, 94	0.64	0.97	1.32	1, 94	0.27
		Herbivores ^e^	0.81	4.04	1, 94	**<0.01**	0.17	6.5	6, 94	**<0.001**	0.82	0.59	6, 94	0.95	0.96	0.69	1, 94	0.63
Pitfall	2011	Predators ^f^	0.93	1.18	1, 94	0.32	0.15	6.05	7, 94	**<0.001**	0.72	0.87	7, 94	0.67	0.95	0.85	1, 94	0.51
	2012	Predators ^f^	0.98	0.2	1, 94	0.95	0.38	3.25	6, 94	**<0.001**	0.69	1.15	6, 94	0.26	0.91	1.74	1, 94	0.13

^a^ Numerator, denominator (error). ^b^ Numbers in bold indicate significant differences at *α* = 0.05. ^c^ Predators include adult hoverflies (Syrphidae), adult lady beetles (Coccinellidae), adult lacewings (Chrysopidae), and pirate bugs (Anthocoridae). ^d^ Parasitoids include parasitic Hymenoptera and tachinid flies (Tachinidae). ^e^ Herbivores include leafhoppers (Cicadellidae), plant bugs (Miridae), weevils (Curculionidae), thrips (Thripidae), and soldier beetles (Cantharidae). ^f^ Predators include crickets (Gryllidae), spiders (Araneae), adult lady beetles (Coccinellidae), adult ground beetles (Carabidae), and adult rove beetles (Staphylinidae).

**Table 2 insects-10-00423-t002:** Results of ANOVA for the effects of methyl salicylate (MeSA) (‘Treatment’), sampling date (‘Date’), ‘Treatment × Date’ interaction, and block on the attraction of syrphids (predators) in 2011 and 2012 and mirids (herbivores) in 2012 to yellow sticky traps in commercial cranberry bogs.

Years	Guilds	Taxa	Variables											
			Treatment			Date			Treatment × Date			Block		
			*F*	*df* ^a^	*P* ^b^	*F*	*df* ^a^	*P* ^b^	*F*	*df* ^a^	*P* ^b^	*F*	*df* ^a^	*P* ^b^
2011	Predators	Syrphidae	18.69	1, 104	**<0.001**	25.04	7, 104	**<0.001**	0.49	7, 104	0.83	0.03	1, 104	0.84
2012	Predators	Syrphidae	34.63	1, 94	**<0.001**	48.45	6, 94	**<0.001**	0.83	6, 94	0.54	9.46	1, 94	**<0.01**
	Herbivores	Miridae	7.87	1, 94	**<0.01**	1.46	6, 94	0.19	0.95	6, 94	0.46	0.43	1, 94	0.51

^a^ Numerator, denominator (error). ^b^ Numbers in bold indicate significant differences at *α* = 0.05.

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
