# Peer review of "Methyl Salicylate Increases Attraction and Function of Beneficial Arthropods in Cranberries"

_insects, 2019, doi:10.3390/insects10120423_

Round 1

Reviewer 1 Report

This paper is well written, providing an adequate review of the current research in this field. By utilizing multiple sampling techniques, including video cameras, the author provides new information on predators and their activity  in this system.  My only criticism is the research was conducted in 2011-12. Why wait 7 years to publish these findings? I do have a few editorial comments/questions. 

Consider including "cranberry" in the title - "...in a Cranberry Agro-Ecosystem"

line 96: Why was there variation in the height of sticky trap placement? (10-20 cm above ground)

line 97: How many pitfall traps were utilized? My assumption is 1 near each sticky trap, but this is not stated and should be clarified.

line 124: The "and/or" should be deleted. 

line 153: "hard-drives" should be singular "hard-drive".

Author Response

Reviewer #1

This paper is well written, providing an adequate review of the current research in this field. By utilizing multiple sampling techniques, including video cameras, the author provides new information on predators and their activity in this system.  My only criticism is the research was conducted in 2011-12. Why wait 7 years to publish these findings? I do have a few editorial comments/questions. 

We understand that the research was conducted several years ago; however, the results remain relevant. The reason for the delay was that it took time for us to process the data and analyze the videos. Dr. Salamanca moved to a new position which also delayed the writing of the paper. The special issue of Insects motivated us to finally complete the analyses of the data and write the paper.

Consider including "cranberry" in the title - "...in a Cranberry Agro-Ecosystem"

We have shortened the title to say “in cranberries”. We have also deleted “The Herbivore-Induced Plant Volatile”.

Line 96: Why was there variation in the height of sticky trap placement? (10-20 cm above ground)

This variation is because the canopy size is variable among cranberry bogs. Thus, there was some variation in the trap placement.

Line 97: How many pitfall traps were utilized? My assumption is 1 near each sticky trap, but this is not stated and should be clarified.

We have indicated that we placed a pitfall trap next to each metal pole (line 98).

Line 124: The "and/or" should be deleted. 

This has been deleted (line 125).

Line 153: "hard-drives" should be singular "hard-drive".

This has been changed (line 153).

Reviewer 2 Report

This clearly written manuscript describes straightforward experiments showing that lures containing methyl salicylate attract higher numbers of natural enemies.  These results should be of interest to a broad audience (entomologists, chemical ecologists, IPM practitioners).  I enjoyed reading it and I have only a few, very minor, suggestions for changes.

Line 101, 4 months does not equal 7-8 weeks.

Lines 120 and following, species names in italics

Paragraph beginning in line 193: I would prefer it if the first paragraph indicated the direction of the statistical test--i.e., the authors should state here that predator numbers were higher in MeSA plots.  In the paragraph beginning in line 2012, the actual numbers of thrips are buried in this paragraph, and I would prefer that they were displayed more prominently

Lines 315 and following-this sentence is awkward.

Author Response

Reviewer #2

This clearly written manuscript describes straightforward experiments showing that lures containing methyl salicylate attract higher numbers of natural enemies.  These results should be of interest to a broad audience (entomologists, chemical ecologists, IPM practitioners).  I enjoyed reading it and I have only a few, very minor, suggestions for changes.

Thank you. 

Line 101, 4 months does not equal 7-8 weeks.

This has been changed. We kept 4 months and deleted 7-8 weeks (line 102).

Lines 120 and following, species names in italics

This has been changed (lines 121 to 126).

Paragraph beginning in line 193: I would prefer it if the first paragraph indicated the direction of the statistical test--i.e., the authors should state here that predator numbers were higher in MeSA plots. In the paragraph beginning in line 201, the actual numbers of thrips are buried in this paragraph, and I would prefer that they were displayed more prominently

We would like to keep the 1st paragraph as is if possible. This is because we first show the multivariate analysis showing that MeSA affects the community of predators. Then, in the 2nd paragraph we indicate that syrphids were attracted by MeSA. Thus, not all predators were affected by MeSA and we believe that saying that predator numbers were higher in MeSA plots might be misleading because only syrphids were affected (even when as a group predators were affected by MeSA). 

Again, for the other paragraph, same thing. We 1st show the effect of MeSA on herbivores as a group and then on individual herbivores. For this reason, thrips are mentioned at the end of the paragraph. We would like to keep this sequence of sentences as it is more logical in our view.

Lines 315 and following-this sentence is awkward.

We have re-written this section with “To date, most field studies on the effects of HIPVs on natural enemies have documented attraction only; in contrast, there are only a few examples on whether this attraction leads to an increase in predation or parasitism (i.e., natural enemy function).” (lines 314 to 317). 

Reviewer 3 Report

Inserted in the attached pdf. 

Author Response

Reviewer #3

Line 2. We have shortened the title and deleted “The Herbivore-Induced Plant Volatile”.

Line 4. We added in the tittle “in cranberries”.

Line 39. We changed “cranberries” for “cranberry”

Line 82. We added the months of the 2-year experiments.

Line 84. We changed V. macrocarpon font to italics.

Line 106. We added the stereomicroscope manufacturer information.

Line 109 – 111. We added a supplementary table (Table S1) with the insecticide information.

Line 112. We would like to keep it as “Egg” as we feel it reads better.

Line 121-126. We have written all scientific names in italics.

Line 141. In both American and British English the correct word is “channel” with double “n”, so we would like to keep as is.

In tables 1 and 2, we have added “s” to Year and Guild (except for Treatment, we would like to keep as is). Also we changed “Taxon” for “Taxa”.

In table 2, we include Miridae as herbivores and discuss that this family are in many cases omnivores. We would like to keep this as is in the text.

Line 243. We would like to keep it as “Egg” as it reads better.

Line 322. We would like to keep as “pteromalid wasps” as it reads better.

Line 356. We added the color of the trap “yellow sticky traps”

Line 367. We added the color of the trap “yellow sticky”
